# Health-promoting work schedules: protocol for a large-scale cluster randomised controlled trial on the effects of a work schedule without quick returns on sickness absence among healthcare workers

Øystein Vedaa [1,2,3,4] Ingebjørg Louise Rockwell Djupedal,[1,4] Erling Svensen,[5] Siri Waage,[6,7] Bjørn Bjorvatn [6,7] Ståle Pallesen,[4,7,8] Stein Atle Lie,[9] Morten Nielsen,[4,10] Anette Harris[4]

For numbered affiliations see end of article.

**Correspondence to**
Dr Øystein Vedaa;
oystein.vedaa@fhi.no

## ABSTRACT

**Introduction** In shift work, quick returns refer to transitions between two shifts with less than 11 hours available rest time. Twenty-three per cent of employees in European countries reported having quick returns. Quick returns are related to short sleep duration, fatigue, sleepiness, work-related accidents and sickness absence. The present study is the first randomised controlled trial (RCT) to investigate the effect of a work schedule without quick returns for 6 months, compared with a work schedule that maintains quick returns during the same time frame.

**Methods and analysis** A parallel-group cluster RCT in a target sample of more than 4000 healthcare workers at Haukeland University Hospital in Norway will be conducted. More than 70 hospital units will be assessed for eligibility and randomised to a work schedule without quick returns for 6 months or continue with a schedule that maintains quick returns. The primary outcome is objective records of sickness absence; secondary outcomes are questionnaire data (n≈4000 invited) on sleep and functioning, physical and psychological health, work-related accidents and turnover intention. For a subsample, sleep diaries and objective sleep registrations with radar technology (n≈ 50) will be collected.

**Ethics and dissemination** The study protocol was approved by the Regional Committee for Medical and Health Research Ethics in Western Norway (2020/200386). Findings from the trial will be disseminated in peer-reviewed journals and presented at national and international conferences. Exploratory analyses of potential mediators and moderators will be reported. User-friendly outputs will be disseminated to relevant stakeholders, unions and other relevant societal groups.

**Trial registration number** NCT04693182.

## INTRODUCTION

An important principle when planning shift schedules is that employees are apportioned sufficient time to rest and recover between shifts. According to the EU's Working Time Directive (2003/88/EC),[1] employees are entitled to minimum 11 hours of rest between two consecutive shifts. Still, in some countries, including Norway, employers and the employees' representatives can agree on rest periods less than 11 hours between two shifts. In this realm, the term *quick return* refers to transitions between two shifts with less than 11 hours available rest time. Quick returns occur most often between an evening shift and a day shift the following day but can also occur between a night shift and an evening shift, and between a day shift and a night shift the subsequent night.[2] In the sixth European Working Conditions Survey published in

### Strengths and limitations of this study

► This is a randomised controlled trial to investigate the effect of a work schedule without quick returns.
► The primary outcome measure is objective register data on sickness absence with no missing data.
► As this is an evaluation of an organisational quality improvement measure implemented for all employees at the hospital, we get to study the effect on the entire target population with full representativeness.
► One concern in this trial is how well the intervention group will succeed in abolishing quick returns from the shift schedule (given that this is a study conducted in a naturalistic setting).
► Another concern in this trial is that a shift schedule that does not include quick returns may unintentionally include other unfavourable shift characteristics that could potentially confound the results (eg, more consecutive evening shifts).

2016,[3] 23% of employees in European countries reported having at least one quick return during the last month. Quick returns seem to be particularly prevalent in the healthcare sector. In a large Danish register survey (n=69 200), it was shown that, on average per year, 65% of nurses, 38% of physicians and 26% of medical secretaries had quick returns in their work schedule.[4]

Eleven hours define the upper limit of potential time for rest between two shifts in a quick return, while the actual time available is often substantially shorter. A Norwegian study investigating payroll data from nurses found that almost 2/3 of the quick returns involved rest time less than 9 hours between two shifts, and some employees (2%) even had rest time of less than 7 hours.[5] The time available for sleep and recuperation is further curtailed by the time it takes to commute to and from work, time for self-care, meals, family obligations and house chores. A systematic literature review reported that sleep duration in quick returns between evening and day shifts typically is reduced to 5–6.5 hours, compared with 7–8 hours on non-quick return nights.[2] In addition to reduced sleep duration, the most robust findings in the literature review were that quick returns were associated with more fatigue, higher levels of sleepiness and shift work disorder (SWD) (ie, sleep problems or sleepiness related to a recurring shift schedule). Individual studies also showed that quick returns were associated with poorer sleep quality, impaired general health and well-being, higher self-reported stress and lower job satisfaction.[2]

The most immediate consequence of quick returns is probably shortened sleep.[6] It is reasonable to think that this in turn leads to a number of other negative consequences. In a diary study (sleep and work schedule), we found that nurses reported higher sleepiness during the day shift when they had quick return to the day shift, as compared with during other regular day shifts.[6] In fact, the results showed that the nurses were as sleepy during the day shift after a quick return as they were during night shifts. It is conceivable that high sleepiness represents a greater problem when it occurs during day shifts than during night shifts, since day shifts are often busier[7] and typically experienced as more stressful.[6] The combination of a high level of sleepiness during a stressful shift might represent a type of circumstance that increases the risk of accidents. Indeed, the association between quick returns and work-related accidents or injuries is established in previous research. In a large register-based study from Denmark, researchers linked payroll data of healthcare workers with national registers of injuries. The results showed that quick returns were associated with a 39% higher risk of injury, compared with having 15–17 hours off between two shifts.[4] A longitudinal study found an increased risk of needlestick injuries among nurses who reported having quick returns as compared with nurses without quick returns.[8] A study based on cross-sectional data found that quick returns were associated with an increased risk of falling asleep at work, of experiencing work-related injuries to themselves, of injuring patients or

others, and of damaging equipment at work.[9] In fact, the risk of experiencing injuries to themselves and damaging equipment at work were greater with quick returns than with night shifts. Another longitudinal study, partly based on the same data, demonstrated that nurses who experienced an increase in the number of quick returns over time also had an increased risk of work-related accidents, whereas a decrease in the number of quick returns over time was associated with reduced risk of accidents.[10]

Over the past 5 years, researchers have increasingly begun to use register/payroll data on exposure to shift work when examining the consequences of different shift characteristics. These data are registered by the employees, typically at healthcare institutions and include information about the date and start and stop time for all shifts performed. In some cases, it is also possible to retrieve data on sickness absence from the same registers. These data comprise information on the date of each day of absence (self-certified and medically certified absence) due to illness. In a Finnish study using such register data from healthcare workers, the relationship between quick returns and short-term sick leave (1 to 3 days) was investigated. The results showed that having few quick returns (defined as three or fewer over a period of 28 days) was associated with a lower risk of short-term sick leave, while having many quick returns (five or more over a period of 28 days) was associated with a higher risk of short-term sickness absence, compared with having no quick returns.[11] In a study based on Danish and Finnish register data, it was found that healthcare workers who had at least 13 quick returns during a year had a higher risk of long-term sick leave than those with fewer quick returns.[12] These findings are in line with results using corresponding register data in Norway.[5] In one study, the findings showed that exposure to quick returns 1 month was associated with a higher risk of sick leave the following month. On average, nurses had three quick returns per month, which corresponded to 21% more sickness absence days the subsequent month (over and above the sickness absence days of workers without quick return).[5]

Research on quick return and health and safety related outcomes have so far all been based on correlational studies. We do not yet know whether these health outcomes are caused by exposure to quick returns. The present study is the first randomised controlled trial (RCT) conducted to determine the effects of abolishing quick return from the work schedule.

## Aims

This paper describes the protocol for a two-arm cluster RCT that assesses the consequences of a shift work schedule abolishing quick returns, compared with a schedule maintaining quick returns for a 6-months period. First, we will examine any differential change in sickness absence (primary outcome) during the 6-month intervention period. Second, we will examine if there are differential changes in sleep and functioning, physical and mental health, work-related accidents and turnover

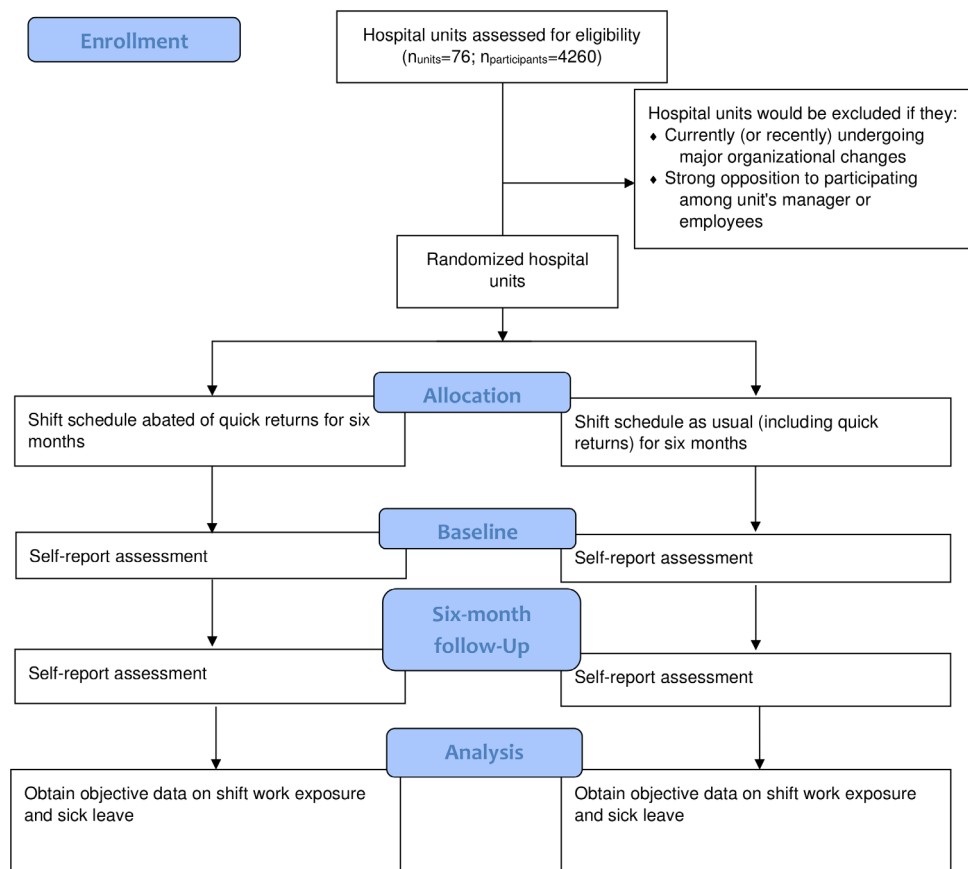

**Figure 1** Flow diagram of timeline for recruitment, randomisation, assessments and for undertaking primary and secondary analyses.

intention, among others (secondary outcomes). Third, we will investigate if individual characteristics associated with shift work tolerance including sex, age, personality and subjectively reported sleep need moderate the negative effects of quick returns on the primary and secondary outcomes. Finally, the study will investigate if individual factors like satisfaction with work schedule, job satisfaction, job engagement and work-family interference moderate the negative effects of quick returns on the primary and secondary outcomes.

## METHODS AND ANALYSIS
The protocol for the current trial follows the Standard Protocol Items: Recommendations for Interventional Trial checklist for intervention trials. The trial is further preregistered with the Clinical Trials website.

Figure 1 shows the flow diagram for the current trial. The flowchart illustrates the timeline for recruitment, randomisation, assessments and for undertaking primary and secondary analyses.

### Research design
A cluster RCT comparing a 6-month work schedule abolishing quick returns (intervention) with that of a 6-month work schedule maintaining a normal amount of quick returns (control) will be conducted. The clusters in this trial represent hospital units that are randomly selected

to receive (or not receive) the intervention. 'Normal amount of quick returns' refer to that which is the common practice at the respective hospital unit in recent years (ie, when no explicit changes have been made to the work schedule), which means that the total number of quick returns at the unit will vary from 329 to 2356 per year (on average, nurses have three quick returns per month at this hospital.[5] In September 2020, the hospital units were informed about the conditions they would be randomised to at the start of the study in 2021. Thus, the autumn of 2020 was spent planning the shift schedule for 2021 (ie, removing quick returns for the intervention group and maintaining quick returns for the control group). Most hospital units started the intervention period in the first half of 2021, while some units started the intervention period in the second half of 2021. The intervention period in this study is six calendar months.

The primary outcome is sickness absence retrieved from the local registers kept by the hospital (including short-term and long-term sick leave). The baseline measurements will be sickness absence from the year preceding the intervention, which for each individual participant will be matched on duration and season to that of the intervention period. We will apply for ethical approval to use the register data from all employees at the randomised hospital units without obtaining individual consent. In addition, a consent-based part of the trial will

be conducted, in which secondary outcome measures will be collected via questionnaire at baseline and 6-month follow-up. All employees (n≈4000) at the randomised units will be asked to complete a digital questionnaire. This will be made available to the employees when they log on to enter their working hours ('MinGat'). Baseline assessment will occur prior to the intervention period, and follow-up assessment will occur towards the end of the intervention period. A subsample (n≈50) will be asked to record their sleep with advanced sleep radar technology (Somnofy)[13] and subjectively with sleep diaries for ≥1 week at the baseline and follow-up assessments, respectively.

## Participants and procedure
### Recruitment
This trial is carried out in collaboration with the human resources department at Haukeland University Hospital, Bergen, Norway. All hospital care units that have 24-hour staffing at Haukeland University Hospital will be considered for inclusion in this trial. This will include all healthcare workers working shifts, except for physicians. Physicians are to be excluded since they often have a different shift schedule and compensation scheme compared with other occupational groups at the hospital. Hereinafter, 'all employees' refer to all healthcare workers engaged in shift work at the randomised hospital units, except for physicians. All employees (n≈4000) at the randomised hospital units will be asked to complete a questionnaire prior to, and at the end of, the intervention period. Recruitment for this part of the trial will take place via the hospital's internal website or through the site in which the employees enter their working hours ('MinGat'). Researchers (the authors of this paper) and human resources personnel at the hospital will attend staff meetings at all included units to inform about the research project and encourage participation. A subsample of n≈50 employees (evenly distributed from the intervention and the control units) will be recruited by convenience for the objective sleep monitoring section of the trial.

### Eligibility
The unit-level inclusion criteria are that the units should have (1) healthcare workers (other than physicians) who work rotating shifts, (2) employees who regularly have quick returns in their work schedule and (3) a new shift rotation year commencing from the first half of 2021 (which is the case for most units at the included hospitals). Exclusion criteria at the unit-level are (1) units recently (or will in the near future) went through other major organisational changes that may confound the results of the trial (this includes during the period from 1 year before the intervention starts until the intervention period is over) or (2) unit's manager or a substantial number of employees strongly oppose participation. Haukeland University Hospital had a total of 76 units that were considered for eligibility.

This trial consists of three different data collections with an expected dissimilar number of participants: (A) a register study, that is, the primary investigation, in which we expect no missing data, (B) a questionnaire study, that is, the secondary investigation, with an expected response rate of 40%–50%[14] and (C) the sleep monitoring study, that is, secondary investigation, conducted on a subsample of ≈50 employees recruited by convenience. All employees from the randomised hospital units working ≥80% of full-time equivalent will participate in the register-based study (investigation A) and the same group will be asked to participate in the questionnaire-based study (investigation B). Finally, participants in the sleep monitoring study (investigation C) will be recruited by convenience from the same sample of healthcare workers requiring that they are working ≥80% of full-time equivalent.

### Randomisation and masking
The randomisation in this trial occurred at the cluster level, in which hospital units constituted the clusters. Hospital units can vary in terms of how much staff they need over the 24 hour day, hence, the work schedule and the occurrence of, for example, quick returns and night shifts can vary across the units. Similar units were, therefore, grouped together based on the fact that they shared some attributes or characteristics. Then a stratified randomisation was performed to the two study conditions in a 1:1 ratio. One subgroup could, for example, consist of units with emergency functions, another with intensive care functions, one with mental healthcare and one with maternity care, etc. In total, we had 10 strata and the sizes of each stratum varied between 2 and 19 hospital units. The randomisation list for each stratum was generated by the online randomisation webpage, www.randomization. com, and the list for each stratum was saved.

It is not possible for participants to be blinded to the group to which they are assigned. However, statistical analyses will be done by a researcher who is masked to group allocation.

### Intervention
The intervention entails implementing a shift schedule, which abolishes quick returns for a 6-month intervention period. The mean number of quick returns in the various hospital units in this trial varies from 3 to 32/year. The intervention means that this number is abolished or reduced as much as possible. For practical reasons, the intervention may be a matter of reducing rather than completely abolishing quick returns. This might be in the case of ensuring adequate staffing (eg, due to sickness absence), and since employees for various reasons may make short-notice shift swaps in which it is not possible to comply with the rule of avoiding quick returns. The human resources department at the hospital will assist shift planners in identifying appropriate shift schedules that do not include quick returns. Table 1 shows some of the examples that were used to show shift planners how this could be done.

**Table 1** Examples of a 2-week cycle of rotating shift work with and without quick returns

| | Week 1 | | | | | | | Week 2 | | | | | | |
|---|---|---|---|---|---|---|---|---|---|---|---|---|---|---|
| | Monday | Tuesday | Wednesday | Thursday | Friday | Saturday | Sunday | Monday | Tuesday | Wednesday | Thursday | Friday | Saturday | Sunday |
| Scenario 1: rotating three-shift **with quick returns** | Day | Day | Night | Night | | | | Evening | Day | Day | | Evening | Day | Evening |
| Scenario 1: rotating three-shift **without quick returns** | Day | Day | Night | Night | | | | Day | Day | Day | | Evening | Evening | Evening |
| Scenario 2: rotating three-shift **with quick returns** | Evening | Day | Day | | Night | Night | Night | | | Evening | Day | Day | | |
| Scenario 2: rotating three-shift **without quick returns** | Day | Evening | Evening | | Night | Night | Night | | | Day | Day | Day | | |
| Scenario 3: weekend shift **with quick returns** | Evening | Day | Day | | Evening | Day | Evening | Evening | Day | Day | Day | | | |
| Scenario 3: weekend shift **without quick returns** | Day | Day | Day | Evening | Day | Evening | Evening | | Day | Day | Evening | | | |
| Scenario 4: rotating two-shift **with quick returns** | Day | Day | Day | Evening | Day | | | Evening | Day | Evening | | Day | | |
| Scenario 4: rotating two-shift **without quick returns** | Evening | Day | Day | Day | Day | | | Evening | Evening | Day | Day | Day | | |

Rotating three-shift refers to a shift schedule in which the workers alternates between day shifts, evening shifts and night shifts. Rotating two-shift refers to a shift schedule in which the workers alternates between only two of the shifts (eg, only working day and evening shifts).

The control condition in this trial implies that employees maintain the same number of quick returns as in previous years for the 6-month intervention period. It is important to note that hospital units in the control group are not expected to experience any increase in the number of quick returns.

## Assessments

All assessments/instruments in this trial are described below. Table 2 provides an overview of the source and timing of the assessments. The primary outcome in this trial is sickness absence (number of days or spells). We will compare the sickness absence in intervention group with the control group during the intervention period, while adjusting for previous sick leave from the corresponding period the year preceding the intervention (matched on duration and season). Other measures included in this trial are secondary outcomes or outcomes used in exploratory or subsidiary analyses.

## Demographics

Demographic information will be obtained both from the register at the hospital as well as from a questionnaire. Information on sex, age and percentage of full-time equivalent will be available from the register data, while information on marital status, highest completed education/degree, years of experience with shift work and if the participant has children living at home will be collected through the questionnaire.

## Primary outcome

*Sickness absence* data will be retrieved from the local records kept by the hospital.[5] This record includes information about the date of any absence of the individual employee, implying that it includes information about both short-term and long-term sickness absence. Furthermore, these data include information on whether the absence is self-certified or whether it is certified by a physician, whether the absence is due to a sick child of whom the employee has childcare responsibility of, and whether the absence is due to COVID-19-related issues (eg, quarantine).

## Secondary outcomes

*The Bergen Insomnia Scale*[15] will be used to measure sleep problems among participants. The scale originally comprised six items that assess symptoms of insomnia. An additional item will be included to the scale in which we will ask about the duration of any sleep problems. This makes it possible to define insomnia according to the diagnostic criteria in the International Classification of Sleep Disorders-Third Edition (ICSD-3),[16] Diagnostic and Statistical Manual of Mental Disorders-Fifth Edition, and the International Classification of Diseases-11th Revision.

SWD will be measured with three standardised questions.[17] SWD was evaluated with three questions based on the criteria from the third edition of the ICSD-3.[16] The questions were: (a) Do you have a work schedule that sometimes overlap with the time you usually sleep?, (b) if yes, does this cause insomnia and/or excessive sleepiness

| Table 2 | Key measures and timing of assessment | | |
|---|---|---|---|
| | | Baseline | 6-month follow-up |
| **Primary outcome** | | | |
| *From hospital register* | | | |
| Sickness absence | | X | X |
| **Secondary outcomes** | | | |
| *Self-reported questionnaires* | | | |
| The Bergen Insomnia Scale | | X | X |
| Shift work disorder | | X | X |
| The Swedish Occupational Fatigue Inventory | | X | X |
| The revised Circadian Type Inventory | | X | |
| The Horne-Östberg Morningness Eveningness Questionnaire | | X | |
| The Hopkins Symptom Checklist-5 | | X | X |
| Job Satisfaction Index | | X | X |
| The Work-Family Interface Scale | | X | X |
| Work-related negative incidents | | X | X |
| The Turnover Intention Scale | | X | X |
| The Utrecht Work Engagement Scale-9 | | X | X |
| Subjective Health Complaints inventory (three of five subscales) | | X | X |
| Recovery Experience Questionnaire (two of four dimensions) | | X | X |
| Epworth Sleepiness Scale | | X | X |
| *Sleep monitoring study (≈50)* | | | |
| Sleep diary (≥7 days) | | X | X |
| Xethru sensor (≥7 days) | | X | X |
| **Additional measures** | | | |
| *Self-reported questionnaires* | | | |
| Unwanted/negative effects | | | X |
| Self-rostering | | X | X |
| Experience of the implementation of the intervention | | | X |
| Physical activity | | X | X |
| Commute time | | X | |
| Sleep duration and perceived need for sleep | | X | X |
| Use of sleep medication and light treatment | | X | X |
| Satisfaction with work schedule | | X | X |
| Preferred presence of quick return in work schedule | | X | X |
| **Demographics and background information** | | | |
| *From hospital register* | | | |
| Sex | | X | |
| Age | | X | |
| Percentage of full-time equivalent | | X | X |
| | | | Continued |

| Table 2 | Continued | | |
|---|---|---|---|
| | | Baseline | 6-month follow-up |
| Payroll data | | X | X |
| *Self-reported questionnaires* | | | |
| Marital status | | X | |
| Highest completed degree | | X | |
| Years of experience with shift work | | X | |
| Children living at home | | X | |

due to reduced amount of sleep?, (c) if yes, has this lasted for at least 3 months? Participants will be classified as having SWD when responding 'yes' to all three questions.

*The revised Swedish Occupational Fatigue Inventory* will be used to measure lack of energy, physical exertion, physical discomfort, lack of motivation and sleepiness.[18] Participants are asked to indicate the extent to which they have recently (or for a specified period of time) experienced a list of 20 psychological and physical sensations related to fatigue.

*The revised Circadian Type Inventory* (rCTI) comprises 11 items, five of which assess flexibility and six assess languidity.[19] High scores on flexibility reflect better ability to sleep and work at odd times, whereas high scores on languidity indicate difficulties overcoming drowsiness and feelings of lethargy following sleep loss.

*The Horne-Östberg Morningness Eveningness Questionnaire* (MEQ) is the most widely used morningness-eveningness inventory[20] and is designed to determine preferred timing of sleep and activities during the 24-hour day.[21] The MEQ reduced version (rMEQ) will be used in the present trial, which is comprised of five items from the original scale.[22]

*Hopkins Symptoms Checklist - 5* (HSCL-5) will be used to measure general psychological distress.[23] HSCL-5 includes five questions about nervousness or inner turmoil, fear or feeling anxious, feeling hopeless about the future, depression or melancholy, worry or restlessness. An average score can be calculated across the five items with values that vary from 1 to 4, in which higher scores indicate a higher degree of psychological distress. The composite score is sometimes recoded into a two-part variable in which a score higher than 2.00 is defined as a high score.

*Job Satisfaction Index* comprises five items measuring satisfaction with work (eg, 'I find real enjoyment in my work').[24] Each item is answered on a 5-point Likert scale, ranging from 1 (strongly disagree) to 5 (strongly agree). Higher scores reflect higher levels of overall job satisfaction.

*The Work Family Interface Scale*[25] will be used to evaluate the four types of work–family spillover. Consisting of 14 items, the scale was designed to measure both negative and positive work-to- family and family-to-work spillover. The responses were graded by a frequency based on a 1–5

Likert scale, with alternatives ranging from never to very often.

*Work-related negative incidents* will be assessed using eight items measuring the number of self-reported work-related accidents, near accidents and dozing off at work or while driving to or from work. These questions have been developed in connection with the Norwegian Survey of Shift work, Sleep and Health among Nurses and have been used in several previous publications.[9]

*The Turnover Intention Scale* will be used to measure turnover intention, which is comprised of three items adapted from Michigan Organisational Assessment Questionnaire.[26] The three items are: 'I will actively look for a new job in the next year,' 'I often think about quitting' and 'I will probably look for a new job by the next year.' Responses were recorded on a 5-point Likert scale from 1 (strongly disagree) to 5 (strongly agree), yielding a score range of 3–15. A high score indicates a high degree of turnover intention.

*The Utrecht Work Engagement Scale* (UWES-9) will be used to measure work engagement.[27] The UWES is originally comprised of 17 items rated on a 7-point scale ranging from 'never' (0) to 'always/every day' (6). The 9-item version of the UWES includes three items for each of the three factors; Vigour (eg, 'At my job, I feel strong and vigorous'), Dedication (eg, 'I am enthusiastic about my job') and Absorption (eg, 'When I am working, I forget everything else around me'). A higher score indicates more work engagement.

*Subjective Health Complaints inventory* (SHC)[28] consists of a list of 29 common health complaints that participants grade the intensity of which they experience each complaint on a four-point scale (0=not at all; 1=a little; 2=some; 3=severe). In this study, we include three of the five subscales; that is, musculoskeletal complaints, pseudoneurological complaints and gastrointestinal complaints.

*Recovery Experience Questionnaire* (REQ)[29] will be used to measure recovery experiences. REQ is originally a 16-item questionnaire with the four subscales psychological detachment, relaxation, mastery and control. The present study includes the subscales of psychological detachment and relaxation. Each item is scored on a 5-point Likert scale ranging from 1 (strongly disagree) to 5 (strongly agree) of which a higher score indicates better detachment/relaxation.

*Epworth Sleepiness Scale* (ESS)[30] will be used to measure participants sleepiness. ESS is an eight-item questionnaire asking the participants how likely they are to doze off or fall asleep in different situations of everyday life including (eg, while sitting and reading, watching TV, when sitting and talking to someone, etc). For each item, participants report the chance of dozing as never (0), slight (1), moderate (2) or high (3) (total score range between 0 and 24). A higher score indicates higher level of sleepiness.

### Additional measures of unwanted/negative effects and other exploratory analyses

Other factors that may have an impact on how the employees react to the intervention will also be investigated. The participants' attitudes to the intervention and the research project will be measured, in addition to how they experience the implementation of the intervention. A set of questions measuring possible negative or unwanted effects of the intervention will be developed for the purpose of this trial. These questions will specifically ask if the changed work schedule has led to disturbed sleep, more stress, worry, depression, overall less time for recovery between work periods, problems in work–family balance, disrupted social relationships, poorer psychosocial climate at work, experience of reduced quality of care offered to patients, etc. For some employees, it is possible that a work schedule that does not allow for quick returns represents a restricted opportunity to codesign their schedule (ie, self-rostering) and reduces the duration of free periods. Therefore, we will measure the participants' perceived change in relation to these parameters. Furthermore, we will include questions about satisfaction with work schedule, commute time, habitual and preferred sleep duration, current use of prescribed or over-the-counter sleep medication, current use of light treatment to improve sleep and participants' physical activity level. Finally, the questionnaire will include an open text box in which participants can write freely, for example, about anything they would like to convey related to the intervention (eg, topics/themes they felt was inadequately addressed in the survey).

*Sleep* will be assessed more thoroughly for a subsample of ≈50 employees. The measures of sleep will include daily self-rating of sleep–wake patterns reported using the consensus sleep diary[17] as well as sleep measured objectively using the Xethru sensor, a low-powered ultra-wideband radar.[31] The sleep registration will occur for ≥7 days at baseline and at 6-month follow-up.

### Sample size

In this trial, all available hospital units at Haukeland University Hospital with healthcare workers who work rotating shifts will be assessed for eligibility. This includes 76 units and 4260 healthcare workers. Based on previous published data,[5] we have calculated that a total of 2028 participants is sufficient to reveal a difference in days of sick leave of 0.9 and 1.25 with an Intra-Class Correlation (ICC) of 0.1 and an average size of the units of 52 (calculation made in: StataCorp. 2015).[32] Thus, with the planned recruitment strategy (ie, invite >70 units and >4000 healthcare workers), we expect to exceed this number and be well within the number of participants required for the primary outcome variable.

### Data analysis plan

All analyses will be conducted based on the intention-to-treat population, unless otherwise stated. To examine the effects of a shift schedule abated of quick returns on

primary and secondary outcomes, the observed rates or scores will be analysed by means of latent growth models (or other equivalent models such as generalised linear mixed models). The observed rates or scores before and during the intervention period will be modelled by a random intercept and a fixed slope. The effect of the intervention will be estimated by using the group variable (intervention vs control) as a predictor of the slope. Between-group effect sizes (Cohen's d) will be calculated by dividing the mean difference in estimated change in scores from baseline to the follow-up assessment by the pooled SD at baseline. Robust maximum likelihood will be used as the estimator, providing unbiased estimates under the assumption of data being missing at random,[33] which might be partly met through the inclusion of baseline scores to the model. The primary outcome measure in this trial is sickness absence data retrieved from the register at the hospital, in which we expect no missing data. However, it is reasonable to expect some missing data on the secondary outcome measures, as data are collected through questionnaire or via the sleep radar and sleep diary.

As some data for the follow-up questionnaire and sleep radar/diary assessment will be missing not at random, the robustness of the results under the missing-at-random assumption will be tested by sensitivity analyses in which the missing scores at follow-up will be replaced by baseline values for each respective individual. Since it is possible to imagine that some participants may experience worsening because of the intervention, we will consider carrying out more rigorous sensitivity analyses. For example, by replacing missing scores at the follow-up assessment with baseline scores multiplied by a given factor (higher or lower than 1.00 depending on the direction that indicates a worsening) in the intervention group and by 1.00 in the control group. These sensitivity analyses will only be performed on selected variables depending on the focus in the respective article.

The intention-to-treat analyses may be accompanied by selected per-protocol analyses in which we, based on payroll data, define a group that has completely abolished or had a satisfactory reduction in the number of quick returns during the intervention period.

The primary outcome of sick leave will mainly be analysed in terms of the total number of sickness absence days and periods (spells) for a given period *before* compared with *during* the intervention period.[5] The models of sickness absence will take into account the zero inflation in this type of data. Other operationalisations of sickness absence might also be considered in accordance with recommendations in the literature.[34] For a further investigation of the sickness absence data, we will consider the use of more complex survival analyses (eg, Cox proportional hazards model), and we will also consider modelling time to return to work (from sickness absence) and/or time before taking sickness absence according to group allocation.

Since the introduction of a work schedule without quick returns may entail an alternative schedule with an increase in other undesirable characteristics (eg, more consecutive evening shifts), we will consider conducting analyses that adjust for such characteristics.

Mediator and moderator analyses will be performed for exploratory purposes, based on the basic principle for such analyses in RCTs as described by others (eg,[35] some of the data collected on demographics, sleep-related personality traits (rCTI and MEQ), mental health, among others, can be used to examine factors that may moderate the impact of the intervention.

### Stakeholder and public involvement

This trial is carried out in close collaboration with the HR department at Haukeland University Hospital. In addition, representatives from all relevant trade unions at the hospital will be involved in the planning and implementation of the research project. The findings of the trial will be disseminated via scholars in terms of scientific paper and conference presentations, and by stakeholder/union advocacy and other relevant public and community groups. Furthermore, Haukeland University Hospital will arrange a conference for other relevant stakeholders, in which research results will be presented, and the implications of the findings will be discussed.

### Patient involvement

No patient involved.

### Ethics and dissemination

The study protocol was approved by the Regional Committee for Medical and Health Research Ethics in Western Norway (2020/200386). In this trial, all employees at the included hospital units will be randomised to one of two conditions, and we will retrieve register data on working hours and sickness absence without collecting individual consent. This poses an ethical dilemma since all participation in research—especially when people are exposed to an intervention—should be consent based. However, the intervention in this trial is to abolish or substantially reduce quick returns, and *not* to increase any exposure. This is, thus, considered not to represent a significant burden on the participants, as the presence of quick returns is already a violation of the Working Environment Act. In addition, we expect that the intervention primarily will have beneficial effects on employees' health and safety. Abolishing or reducing the number of quick returns is a quality improvement measure that the Health Trust wants to implement independently of the present research project. The fact that the intervention is carried out as a research project is considered an advantage for the employees, as far as we are able to uncover any unintended negative effects of the intervention and further to be able to empirically document potential benefits on health and safety.

The result of this trial will potentially impact subsequent standards and practice when it comes to planning shift schedules and their compliance with the Working Environment Act. As vast number of employees might be

affected by the trial results, it is equally important that the results are representative of the employees. We believe that this justifies the use of the employees' register data without obtaining individual consent.

Participants will be required to provide informed consent before participating in the questionnaire and sleep diary/radar part of the trial (see online supplemental files 1 and 2, respectively). The recruitment and consent process emphasises that participation is voluntary and that participants can withdraw from this part of the trial at any time point without any consequences. Self-report data are recorded in electronic files that are encrypted and password protected. No identifying information will be stored alongside the self-report data. Furthermore, only researchers directly involved in data analysis will be granted supervised access to deidentified participant data.

Findings from this RCT will be disseminated in peer-reviewed publications and as conference presentations. After the research project is completed, Haukeland University Hospital will arrange a conference for stakeholders where the results and experience from the research will be disseminated and discussed.

## DISCUSSION

To the best of our knowledge, this is the first RCT to investigate the effect of a work schedule abolishing quick returns. Previous research on quick returns has been dominated by cross-sectional studies and a few longitudinal investigations. Although quick returns have consistently been associated with negative health and safety outcomes, it is unclear whether quick returns are the cause of these negative outcomes. This trial will, thus, be the first sincere attempt to establishing such a causal relationship.

There are several major strengths to this trial. The intervention is carried out in all eligible hospital units at Haukeland University Hospital, in which we retrieve objective register data (notably with no missing data) on the primary outcome measure—sickness absence. Hence, reporting bias such as social desirability and memory biases will be avoided. This study is unique as it will imply complete access to the entire target population, also including individuals who typically choose not to participate in such studies. Hence, this ensures full representativeness, strengthening the external validity of the study. Furthermore, we have access to objective data on exposure to shift work (quick returns and other shift characteristics) during the intervention period. This provides us the opportunity to accurately assess compliance with the intervention and the true reduction in quick returns that occur as well as monitoring other systemic differences that might occur in the shift schedule between the two parallel conditions. It is also an asset that we combine objective data with data collected via questionnaire. This provides us the opportunity to study the effect of abolishing or reducing quick returns on sleep, health and

safety as well as being able, for example, to study potential moderators to any effects we observe.

There are also some possible limitations with this trial that should be mentioned. The trial is conducted in a naturalistic setting which does not allow for the same strict control as generally would be preferred in experimental designs. One main concern is how well the intervention group will succeed in abolishing quick returns from the shift schedule. We expect that, for many individuals, it will be a matter of reducing the number of quick returns, rather than complete abolition, for example, since such shift transitions occasionally may be necessary to ensure adequate staffing. Another concern is that a shift schedule that does not include quick returns may unintentionally include other unfavourable shift characteristics that could potentially confound the results. However, during the implementation of the trial, shift planners are provided with recommendations on how to set up shift schedules without quick returns, for example, avoiding backward shift rotations, which, as far as possible, avoids other unfavourable shift characteristics. Furthermore, for the participants in this trial, it will be obvious which study condition they have been allocated to, thus, their expectations can potentially have an impact on results based on self-reported data.[36] A questionnaire was used to measure most secondary outcome variables in this trial. An important limitation with such subjective reports is possible bias related to the validity of the instruments and recall bias.[37] However, most of the variables were based on standardised questionnaires with adequate psychometric properties. Furthermore, most variables are subjective by their very nature and need accordingly to be measured with self-reports.

If a shift schedule without quick returns is shown to be associated with less sickness absence or positive effects on other outcomes compared with a control group, this may encourage a stricter compliance with the workers' right to have at least 11 hours off between two subsequent shifts. The results of this trial will provide valuable information to stakeholders (nurses responsible for developing shift schedules, trade unions, politicians and innovators) about the effect of quick returns and individual tolerance to quick returns.

**Author affiliations**
[1]Department of Health Promotion, Norwegian Institute of Public Health, Bergen, Norway
[2]Department of Mental Health, Norwegian University of Science and Technology, Trondheim, Norway
[3]Department of Research and Development, St Olavs University Hospital, Trondheim, Norway
[4]Department of Psychosocial Science, University of Bergen, Bergen, Norway
[5]Department of Human Resources, Haukeland University Hospital, Bergen, Norway
[6]Department of Global Public Health and Primary Care, University of Bergen, Bergen, Norway
[7]Norwegian Competence Center for Sleep Disorders, Haukeland University Hospital, Bergen, Norway
[8]Optentia at the Vaal Triangle Campus of the North-West University, Vanderbijlpark, South Africa
[9]Department of Clinical Dentistry, University of Bergen, Bergen, Norway

[10]Department of Work Psychology and Physiology, National Institute of Occupational Health, Oslo, Norway

**Acknowledgements**  We would like to thank Ljiljana Djuric-Rakovic and John Olav Larssen at Haukeland University Hospital for their invaluable help in setting up and distributing the electronic questionnaires for this study. We would also like to thank Helga Berdal Lorentzen and Ole-Daniel Tuft Virkesdal at the HR department at Haukeland University Hospital, and employee representatives of the Norwegian Nurses Organisation, Trade Union Delta, the joint organisation for Child Welfare Educators, Social Workers and Learning Disability Nurse and others trade unions for their support and contribution in the implementation of this research project. We would also like to thank Lukas Krondorf at Vital Things AS for technical support during the registration of nurses' sleep using radar technology.

**Contributors**  AH, ØV, SP, BB, SW, SAL, ES and MN conceived the study. ØV and ILRD produced the first draft of the manuscript. All authors assisted in drafting of the final, submitted version of manuscript and all authors have approved this version.

**Funding**  The study was funded from The Research Council of Norway (303671) and the University of Bergen, Bergen, Norway.

**Disclaimer**  The sponsors had no role in a study design, collection, management, analysis and interpretation of data; writing of the report; and the decision to submit the report for publication, including whether they will have ultimate authority over any of these activities. The sponsors had no authority over any of the above activities.

**Competing interests**  None declared.

**Patient consent for publication**  Not applicable.

**Provenance and peer review**  Not commissioned; externally peer reviewed.

**ORCID iDs**
Øystein Vedaa http://orcid.org/0000-0003-2719-2375
Bjørn Bjorvatn http://orcid.org/0000-0001-7051-745X

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
