## [Reviewer comments · BMJ Open]

ARTICLE DETAILS

TITLE (PROVISIONAL)	Health promoting work schedules: protocol for a large-scale cluster randomized controlled trial on the effects of a work schedule without quick returns on sickness absence among healthcare workers
AUTHORS	Vedaa, Oystein; Djupedal, Ingebjørg Louise Rockwell; Svensen, Erling; Waage, Siri; Bjorvatn, Bjørn; Pallesen, Ståle; Lie, S; Nielsen, Morten; Harris, Anette

VERSION 1 – REVIEW

REVIEWER	López-Bueno, Rubén University of Zaragoza
REVIEW RETURNED	19-Nov-2021

GENERAL COMMENTS	This is an interesting study protocol trying to address an important topic, which is high percentages of sickness absence among the healthcare sector. Overall, it is both well-written and documented, although some improvements could be made, particularly better describing the randomization process, better examining literature on the topic, and the limitations of the study. Methods -I am missing some point regarding how the randomization is going to be made with your sample. That will be cluster randomized, but more details on how this randomization is going to be conducted (eg. seed and so on)-A DAG would be really useful to better depict and address the potential confounding bias that might affect the association between your two study variables. For instance, it is well-known that leisure-time physical activity can reduce the risk of sickness absence among healthcare workers. Moreover, other health-related behaviours such as smoking might also affect this.-Regarding your statistical analyses I encourage you to use an adjusted Cox regression model, if possible with competing risk.-Ethical concerns are still not clear. You state that your intervention will reduce quick returns in those selected units. Does it mean that all the examined units are experiencing quick returns. Discussion -The limitations of the study should be better defined. For instance, since most of your data stems from questionnaires, a recall bias is plausible. Also, unmeasured confounding should be always considered, and DAG or E-values can give a better idea on this.
---

REVIEWER	MADAN, IRA GUYS AND ST THOMAS NHS FOUNDATION TRUST
-----------------	---

REVIEW RETURNED	10-Dec-2021
GENERAL COMMENTS	This trial seeks to answer an important question. My main concern is about the sample size calculation. The authors didn't estimate an ICC nor do they give an estimate of the number of clusters required. Furthermore they do not state what their estimated effect size of the intervention was. This section of the protocol needs to be made clearer.

VERSION 1 – AUTHOR RESPONSE

Dr. Rubén López-Bueno, University of Zaragoza

Comments to the Author:

This is an interesting study protocol trying to address an important topic, which is high percentages of sickness absence among the healthcare sector. Overall, it is both well-written and documented, although some improvements could be made, particularly better describing the randomization process, better examining literature on the topic, and the limitations of the study.

Methods

-I am missing some point regarding how the randomization is going to be made with your sample. That will be cluster randomized, but more details on how this randomization is going to be conducted (eg. seed and so on)

Thank you so much for this comment! We agree that the randomization in this study was not properly explained, and that the text in addition contained some incorrect information. We have now revised this section to correct these issues and to be clearer on how the randomization took place. Below we have written the text that has been added to the manuscript under the heading Randomization. We have not considered it appropriate to report (or save) the seed that was used. We have instead saved the randomization list for each stratum that was randomized. We have also reported this under the Randomization section of the manuscript:

"The randomization in this trial occurred at the cluster level, in which hospital units constituted the clusters. As shown in Figure 1, a total of 67 hospital units were randomized. Hospital units can vary in terms of how much staff they need over the 24-hour day, hence, the work schedule and the occurrence of, for example, quick returns and night shifts can vary across the units. Similar units were therefore grouped together, and block randomized to the two study conditions in a 1:1 ratio. One stratum could, for example, consist of units with emergency functions, with intensive care functions, with mental health care, with maternity care, etc. In total we had 10 strata and the sizes of each

stratum varied between 2 and 19 hospital units. The randomization list for each stratum was generated by the online randomization webpage, www.randomization.com, and the list for each stratum was saved."

-A DAG would be really useful to better depict and address the potential confounding bias that might affect the association between your two study variables. For instance, it is well-known that leisure-time physical activity can reduce the risk of sickness absence among healthcare workers. Moreover, other health-related behaviours such as smoking might also affect this.

Thank you so much for this comment. As far as we understand, DAG is useful when you have confounding variables (colliders, back-doors etc). In an RCT, like the present trial, randomization will normally adjust for this. In this trial, we randomize at the cluster level and we have close to 70 randomized hospital units. We know that some hospital units are more similar than others in terms of the need for 24-hour staffing and thus the shift schedules regime they use. Therefore, we grouped together units that were similar in this regard and defined these as stratum that we randomized within. We believe this further helped to ensure that there were no systematic differences between the two study conditions. Based on this, we see it as unnecessary with further adjustment for variables such as physical activity and smoking in this study.

-Regarding your statistical analyses I encourage you to use an adjusted Cox regression model, if possible with competing risk.

Thank you for this constructive suggestion. The reviewer suggests that we use an adjusted model, but as explained in the response to the previous comment, we do not see the need for further adjustment in the analyses since the randomization will ensure equal groups in the two conditions. However, what we will consider adjusting for in the analyses is, for example, the number of evening shifts in a row, as explained in the paper (see the Data analysis plan), since a shift schedule without quick return in some cases may lead to an increase in the number of evening shifts in a row. Furthermore, we agree that Cox regression may be a relevant approach to conduct more thorough analyses of the sickness absence data we have available. In our Data analysis plan we have already described that we will consider running more complex survival analyses (e.g., Cox proportional hazards model), and we will also consider modelling time to return to work (from sickness absence) and/or time before taking sickness absence according to group allocation. However, we want to leave the description of the analysis plan as it is, where our primary analyses in the RCT design will be the latent growth models (or other equivalent models such as generalized linear mixed models).

-Ethical concerns are still not clear. You state that your intervention will reduce quick returns in those selected units. Does it mean that all the examined units are experiencing quick returns.

Thank you for this comment. Yes, in this trial, the first two unit-level inclusion criteria were that the units should have "1) healthcare workers (other than physicians) who

work rotating shifts, 2) employees who regularly have quick returns in their work schedule” (please see under the heading: Participant and procedure: Eligibility). This means that all the units that were considered relevant for this trial have employees who work rotating shifts and regularly experience quick returns in their shift schedule. As reported in the paper, the number of quick returns across the units varied between 329 and 2356 per year, and on average at this hospital, nurses have three quick returns per month (please see under the heading: Research design). We are not sure what more information the reviewer wants us to include on this matter. We also think that it will be redundant to repeat the text above in the Ethics section.

Discussion

-The limitations of the study should be better defined. For instance, since most of your data stems from questionnaires, a recall bias is plausible. Also, unmeasured confounding should be always considered, and DAG or E-values can give a better idea on this.

Thank you very much for this comment. We agree that a potential bias in this trial regarding the secondary outcome measures is that they are mostly based on subjective reporting, which involves a risk of recall bias. We have now included the following sentence in the limitations section to be clear on that:

“A questionnaire was used to measure most secondary outcome variables in this trial. An important limitation with such subjective reports is possible bias related to the validity of the instruments and recall bias.³⁷ However, most of the variables were based on standardized questionnaires with adequate psychometric properties. Furthermore, most variables are subjective by their very nature and need accordingly to be measured with self-reports.”

Dr. IRA MADAN, GUYS AND ST THOMAS NHS FOUNDATION TRUST

Comments to the Author:

This trial seeks to answer an important question.

My main concern is about the sample size calculation. The authors didn't estimate an ICC nor do they give an estimate of the number of clusters required. Furthermore they do not state what their estimated effect size of the intervention was. This section of the protocol needs to be made clearer.

Thank you so much for this comment. We agree that the description of the sample size was somewhat unclear in our first submission of the paper. We have now updated the entire section and have done our best to be more precise when it comes to sample size. The fact of the matter is that in this trial we include all possible hospital units with the number of participants/workers available from these units. Our power calculations show that this should be sufficient with sick leave as the primary outcome measure. We have updated the text in the manuscript so that it now reads as follows:

“In this trial, all available hospital units with healthcare workers who work rotating shifts were assessed for eligibility. This included 76 units and 4260 healthcare workers. As shown in Figure 1, a total of 67 of these units were finally included, i.e. 3669 healthcare workers. Based on previous published data (Vedaa et al. 2017) we have calculated that a total of 2028 participants is sufficient to reveal a difference in days of sick leave of 0.9 and 1.25 with an ICC of 0.1 and an average size of the units of 52 workers (calculation made in: StataCorp. 2015. Stata Statistical Software:

Release 14. College Station, TX: StataCorp LP). With 67 hospital units and 3669 participants, we will thus be well within the number of participants required for the primary outcome variable and we consider this sufficient for all conceivable purposes of this trial.”

VERSION 2 – REVIEW

REVIEWER	López-Bueno, Rubén University of Zaragoza
REVIEW RETURNED	19-Jan-2022
GENERAL COMMENTS	The authors addressed well my queries.